# Sampling Networks and Aggregate Simulation for Online POMDP Planning

**Hao Cui**
Department of Computer Science
Tufts University
Medford, MA 02155, USA
`hao.cui@tufts.edu`

**Roni Khardon**
Department of Computer Science
Indiana University
Bloomington, IN, USA
`rkhardon@iu.edu`

## Abstract

The paper introduces a new algorithm for planning in partially observable Markov decision processes (POMDP) based on the idea of aggregate simulation. The algorithm uses product distributions to approximate the belief state and shows how to build a representation graph of an approximate action-value function over belief space. The graph captures the result of simulating the model in aggregate under independence assumptions, giving a symbolic representation of the value function. The algorithm supports large observation spaces using sampling networks, a representation of the process of sampling values of observations, which is integrated into the graph representation. Following previous work in MDPs this approach enables action selection in POMDPs through gradient optimization over the graph representation. This approach complements recent algorithms for POMDPs which are based on particle representations of belief states and an explicit search for action selection. Our approach enables scaling to large factored action spaces in addition to large state spaces and observation spaces. An experimental evaluation demonstrates that the algorithm provides excellent performance relative to state of the art in large POMDP problems.

## 1  Introduction

Planning in partially observable Markov decision processes is a central problem in AI which is known to be computationally hard. Work over the last two decades produced significant algorithmic progress that affords some scalability for solving large problems. Off-line approaches, typically aiming for exact solutions, rely on the structure of the optimal value function to construct and prune such representations [22, 10, 2], and PBVI and related algorithms (see [20]) carefully control this process yielding significant speedup over early algorithms. In contrast, online algorithms interleave planning and execution and are not allowed sufficient time to produce an optimal global policy. Instead they focus on search for the best action for the current step. Many approaches in online planning rely on an explicit search tree over the belief space of the POMDP and use sampling to reduce the size of the tree [11] and most effective recent algorithms further use a particle based representation of the belief states to facilitate fast search [21, 25, 23, 7].

Our work is motivated by the idea of aggregate simulation in MDPs [5, 4, 3]. This approach builds an explicit symbolic computation graph that approximates the evolution of the distribution of state and reward variables over time, conditioned on the current action and future rollout policy. The algorithm then optimizes the choice of actions by gradient based search, using automatic differentiation [8] over the explicit function represented by the computation graph. As recently shown [6] this is equivalent to solving a marginal MAP inference problem where the expectation step is evaluated by belief propagation (BP) [17], and the maximization step is performed using gradients.

We introduce a new algorithm SNAP (Sampling Networks and Aggregate simulation for POMDP) that expands the scope of aggregate simulation. The algorithm must tackle two related technical challenges. The solution in [5, 4] requires a one-pass forward computation of marginal probabilities. Viewed from the perspective of BP, this does not allow for downstream observations – observed descendents of action variables – in the corresponding Bayesian network. But this conflicts with the standard conditioning on observation variables in belief update. Our proposed solution explicitly enumerates all possible observations, which are then numerical constants, and reorders the computation steps to allow for aggregate simulation. The second challenge is that enumerating all possible observations is computationally expensive. To resolve this, our algorithm must use explicit sampling for problems with large observation spaces. Our second contribution is a construction of sampling networks, showing how observations $z$ can be sampled symbolically and how both $z$ and $p(z)$ can be integrated into the computation graph so that potential observations are sampled correctly for any setting of the current action. This allows full integration with gradient based search and yields the SNAP algorithm.

We evaluate SNAP on problems from the international planning competition (IPC) 2011, the latest IPC with publicly available challenge POMDP problems, comparing its performance to POMCP [21] and DESPOT [25]. The results show that the algorithm is competitive on a large range of problems and that it has a significant advantage on large problems.

## 2 Background

### 2.1 MDPs and POMDPs

A MDP [18] is specified by $\{\mathbb{S}, \mathbb{A}, T, R, \gamma\}$, where $\mathbb{S}$ is a finite state space, $\mathbb{A}$ is a finite action space, $T(s, a, s') = p(s'|s, a)$ defines the transition probabilities, $R(s, a)$ is the immediate reward and $\gamma$ is the discount factor. For MDPs (where the state is observed) a policy $\pi : \mathbb{S} \to \mathbb{A}$ is a mapping from states to actions, indicating which action to choose at each state. Given a policy $\pi$, the value function $V^\pi(s)$ is the expected discounted total reward $E[\sum_i \gamma^i R(s_i, \pi(s_i)) \mid \pi]$, where $s_i$ is the $i^{th}$ state visited by following $\pi$ and $s_0 = s$. The action-value function $Q^\pi : \mathbb{S} \times \mathbb{A} \to \mathcal{R}$ is the expected discounted total reward when taking action $a$ at state $s$ and following $\pi$ thereafter.

In POMDPs the agent cannot observe the state. The MDP model is augmented with an observation space $O$ and the observation probability function $O(z, s', a) = p(z|s', a)$ where $s'$ is the state reached and $z$ is the observation in the transition $T(s, a, s')$. That is, in the transition from $s$ to $s'$, observation probabilities depend on the next state $s'$. The *belief state*, a distribution over states, provides a sufficient statistic of the information from the initial state distribution and history of actions and observations. The belief state can be calculated iteratively from the history. More specifically, given the current belief state $b_t(s)$, action $a_t$ and no observations, we expect to be in

$$b_{t+1}^{a_t}(s') = p(s'|b_t, a_t) = E_{s \sim b_t(s)}[p(s'|s, a_t)]. \tag{1}$$

Given $b_t(s)$, $a_t$ and observation $z_t$ the new belief state is $b_{t+1}^{a_t, z_t}(s'') = p(s''|b_t, a_t, z_t)$:

$$b_{t+1}^{a_t, z_t}(s'') = \frac{p(s'', z_t|b_t, a_t)}{p(z_t|b_t, a_t)} = \frac{b_{t+1}^{a_t}(s'')p(z_t|s'', a_t)}{p(z_t|b_t, a_t)} \tag{2}$$

where the denominator in the last equation requires a double sum over states: $\sum_s \sum_{s'} b_t(s)p(s'|s, a_t)p(z_t|s', a_t)$. Algorithms for POMDPs typically condition action selection either directly on the history or on the belief state. The description above assumed an atomic representation of states, actions and observations. In factored spaces each of these is specified by a set of variables, where in this paper we assume the variables are binary. In this case, the number of states (actions, observations) is exponential in the number of variables, implying that state enumeration which is implicit above is not feasible. One way to address this challenge is by using a particle based representation for the belief state as in [20, 21]. In contrast, our approach approximates the belief state as a product distribution which allows for further computational simplifications.

### 2.2 MDP planning by aggregate simulation

Aggregate simulation follows the general scheme of the rollout algorithm [24] with some modifications. The core idea in aggregate simulation is to represent a distribution over states at every step of planning. Recall that the rollout algorithm [24] estimates the state-action value function $Q^\pi(s, a)$ by

applying $a$ in $s$ and then simulating forward using action selection with $\pi$, where the policy $\pi$ maps states to actions. This yields a trajectory, $s, a, s_1, a_1, \ldots$ and the average of the cumulative reward over multiple trajectories is used to estimate $Q^\pi(s, a)$. The lifted-conformant SOGBOFA algorithm of [3] works in factored spaces. For the rollout process, it uses an open-loop policy (a.k.a. a straight line plan, or a sequential plan) where the sequence of actions is pre-determined and the actions used do not depend on the states visited in the trajectory. We refer to this below as a sequential rollout plan $p$. In addition, instead of performing explicit simulations it calculates a product distribution over state and reward variables at every step, conditioned on $a$ and $p$. Finally, while rollout uses a fixed $\pi$, lifted-conformant SOGBOFA optimizes $p$ at the same time it optimizes $a$ and therefore it can improve over the initial rollout scheme. In order to perform this the algorithm approximates the corresponding distributions as product distributions over the variables.

SOGBOFA accepts a high level description of the MDP, where our implementation works with the RDDL language [19], and compiles it into a computation graph. Consider a Dynamic Bayesian Network (DBN) which captures the finite horizon planning objective conditioned on $p$ and $a$. The conditional distribution of each state variable $x$ at any time step is first translated into a disjoint sum form "if$(c_1)$ then $p_1$, if$(c_2) \ldots$ if$(c_n)$ then $p_n$" where $p_i$ is $p(x{=}T)$, $T$ stands for *true*, and the conditions $c_i$ are conjunctions of parent values which are are *mutually exclusive and exhaustive*. The last condition implies that the probability that the variable is true is equal to: $\sum p(c_i)p_i$. This representation is always possible because we work with discrete random variables and the expression can be obtained from the conditional probability of $x$ give its parents. In practice the expressions can be obtained directly from the RDDL description. Similarly the expected value of reward variables is translated into a disjoint sum form $\sum p(c_i)v_i$ with $v_i \in \mathbb{R}$. The probabilities for the conditions $c_i$ are approximated by assuming that their parents are independent, that is $p(c_i)$ is approximated by $\hat{p}(c_i) = \prod_{w_k \in c_i} \hat{p}(w_k) \prod_{\bar{w}_k \in c_i} (1 - \hat{p}(w_k))$, where $w_k$ and $\bar{w}_k$ are positive and negative literals in the conjunction respectively. To avoid size explosion when translating expressions with many parents, SOGBOFA skips the translation to disjoint sum form and directly translates from the logical form of expressions into a numerical form using standard translation from logical to numerical constructs ($a \wedge b$ is $a * b$, $a \vee b$ is 1-(1-$a$)*(1-$b$), $\neg a$ is 1-$a$). These expressions are combined to build an explicit computation graph that approximates of the marginal probability for every variable in the DBN.

To illustrate this process consider the following example from [4] with three state variables s(1), s(2) and s(3), three action variables a(1), a(2), a(3) and two intermediate variables $cond1$ and $cond2$. The MDP model is given by the following RDDL [19] program where primed variants of variables represent the value of the variable after performing the action.

```
cond1 = Bernoulli(0.7)
cond2 = Bernoulli(0.5)
s'(1) = if (cond1)  then ~a(3)  else false
s'(2) = if (s(1))   then a(2)   else false
s'(3) = if (cond2)  then s(2)   else false
reward = s(1) + s(2) + s(3)
```

The model is translated into disjoint sum expressions as s'(1) = (1-a(3))*0.7, s'(2) = s(1)*a(2), s'(3) = s(2) * 0.5, r = s(1) + s(2) + s(3). The corresponding computation graph, for horizon 3, is shown in the right portion of Figure 1. The bottom layer represents the current state and action variables. In the second layer action variables represent the conformant policy, and state variables are computed from values in the first layer where each node represents the corresponding expression. The reward variables are computed at each layer and summed to get the cumulative $Q$ value. The graph enables computation of $Q^p(s, a)$ by plugging in values for $p, s$ and $a$. For the purpose of our POMDP algorithm it is important to notice that the computation graph in SOGBOFA *replaces each random variable in the graph with its approximate marginal probability*.

Now given that we have an explicit computation graph we can use it for optimizing $a$ and $p$ using gradient based search. This is done by using automatic differentiation [8] to compute gradients w.r.t. all variables in $a$ and $p$ and using gradient ascent. To achieve this, for each action variable, e.g., $a_{t,\ell}$, we optimize $p(a_{t,\ell}{=}T)$, and optimize the joint setting of $\prod_\ell p(a_{t,\ell}{=}T)$ using gradient ascent.

SOGBOFA includes several additional heuristics including dynamic control of simulation depth (trying to make sure we have enough time for $n$ gradient steps, we make the simulation shallower if graph size gets too large), dynamic selection of gradient step size, maintaining domain constraints, and a balance between gradient search and random restarts. In addition, the graph construction simplifies obvious numerical operations (e.g., $1 * x = x$ and $0 * x = 0$) and uses dynamic programming to avoid regenerating identical node computations, achieving an effect similar to lifting in probabilistic

inference. All these heuristics are inherited by our POMDP solver, but they are not important for understanding the ideas in this paper. We therefore omit the details and refer to reader to [4, 3].

## 3  Aggregate simulation for POMDP

This section describes a basic version of SNAP which assumes that the observation space is small and can be enumerated. Like SOGBOFA, our algorithm performs aggregate rollout with a rollout plan $p$. The estimation is based on an appropriate definition of the $Q()$ function over belief states:

$$Q^p(b_t, a_t) = E_{b_t(s)}[R(s, a_t)] + \sum_{z_t} p(z_t|b_t, a_t)\, V^{p^{z_t}}(b_{t+1}^{a_t,z_t}) \tag{3}$$

where $V^p(b)$ is the cumulative value obtained by using $p$ to choose actions starting from belief state $b$. Notice that we use a different rollout plan $p^{z_t}$ for each value of the observation variables which can be crucial for calculating an informative value for each $b_{t+1}^{a_t,z_t}$. The update for belief states was given above in Eq (1) and (2). Our algorithm implements approximations of these equations by assuming factoring through independence and by substituting variables with their marginal probabilities.

A simple approach to upgrade SOGBOFA to this case will attempt to add observation variables to the computation graph and perform the calculations in the same manner. However, this approach does not work. Note that observations are descendants of current state and action variables. However, as pointed out by [6] the main computational advantage in SOGBOFA results from the fact that there are no downstream observed variables in the computation graph. As a result belief propagation does not have backward messages and and the computation can be done in one pass. To address this difficulty we reorder the computations by grounding all possible values for observations, conditioning the computation of probabilities and values on the observations and combining the results.

We start by enforcing factoring over the representation of belief states:

$$\hat{b}_t(s) = \prod_i \hat{b}_t(s_i); \quad \hat{b}_{t+1}^{a_t}(s) = \prod_i \hat{b}_{t+1}^{a_t}(s_i); \quad \hat{b}_{t+1}^{a_t,z_t}(s) = \prod_i \hat{b}_{t+1}^{a_t,z_t}(s_i)$$

We then approximate Eq (1) as

$$b_{t+1}^{a_t}(s_i'{=}T) = E_{s\sim b_t(s)}[p(s_i'{=}T|s, a_t)] \approx \hat{b}_{t+1}^{a_t}(s_i'{=}T) = \hat{p}(s_i'{=}T|\{\hat{b}_t(s_i)\}, a_t)$$

where the notation $\hat{p}$ indicates that conditioning on the factored set of beliefs $\{\hat{b}_t(s_i)\}$ is performed by replacing each occurrence of $s_j$ in the expression for $p(s_i'{=}T|\{s_j\}, a_t)$ with its marginal probability $\hat{b}_t(s_j{=}T)$. We use the same notation with intended meaning for substitution by marginal probabilities when conditioning on $\hat{b}$ in other expressions below. Note that since variables are binary, for any variable $x$ it suffices to calculate $\hat{p}(x{=}T)$ where $1 - \hat{p}(x{=}T)$ is used when the complement is needed. We use this implicitly in the following. Similarly, the reward portion of Eq (3) is approximated as

$$E_{b_t(s)}[R(s, a_t)] \approx \hat{R}(\{\hat{b}_t(s_i)\}, a_t). \tag{4}$$

The term $p(z_t|b_t, a_t)$ from Eq (2) and (3) is approximated by enforcing factoring as $p(z_t|b_t, a_t) \approx \prod_k p(z_{t,k}|b_t, a_t)$ where for each factor we have

$$p(z_{t,k}{=}T|b_t, a_t) = E_{b_{t+1}^{a_t}(s')}[p(z_{t,k}{=}T|s', a_t)] \approx \hat{p}(z_{t,k}{=}T|b_t, a_t) = \hat{p}(z_{t,k}{=}T|\{\hat{b}_{t+1}^{a_t}(s_i')\}, a_t).$$

Next, to facilitate computations with factored representations, we replace Eq (2) with

$$b_{t+1}^{a_t,z_t}(s_i''{=}T) = \frac{p(s_i''{=}T, z_t|b_t, a_t)}{p(z_t|b_t, a_t)} = \frac{b_{t+1}^{a_t}(s_i''{=}T)p(z_t|s_i''{=}T, b_t, a_t)}{p(z_t|b_t, a_t)}. \tag{5}$$

Notice that because we condition on a single variable $s_i''$ the last term in the numerator must retain the conditioning on $b_t$. This term is approximated by enforcing factoring $p(z_t|s_i''{=}T, b_t, a_t) \approx \prod_k p(z_{t,k}|s_i''{=}T, b_t, a_t)$ where each component is

$$p(z_{t,k}{=}T|s_i''{=}T, b_t, a_t) = E_{b_{t+1}^{a_t}(s''|s_i''{=}T)}[p(z_{t,k}{=}T|s'', a_t)] \approx \hat{p}(z_{t,k}{=}T|s_i''{=}T, \{\hat{b}_{t+1}^{a_t}(s_\ell'')\}_{\ell\neq i}, a_t)$$

and Eq (5) is approximated as:

$$\hat{b}_{t+1}^{a_t,z_t}(s_i''{=}T) = \frac{\hat{b}_{t+1}^{a_t}(s_i''{=}T)\, \prod_k \hat{p}(z_{t,k}|s_i''{=}T, \{\hat{b}_{t+1}^{a_t}(s_\ell'')\}_{\ell\neq i}, a_t)}{\prod_k \hat{p}(z_{t,k}|\{\hat{b}_{t+1}^{a_t}(s_i')\}, a_t)}. \tag{6}$$

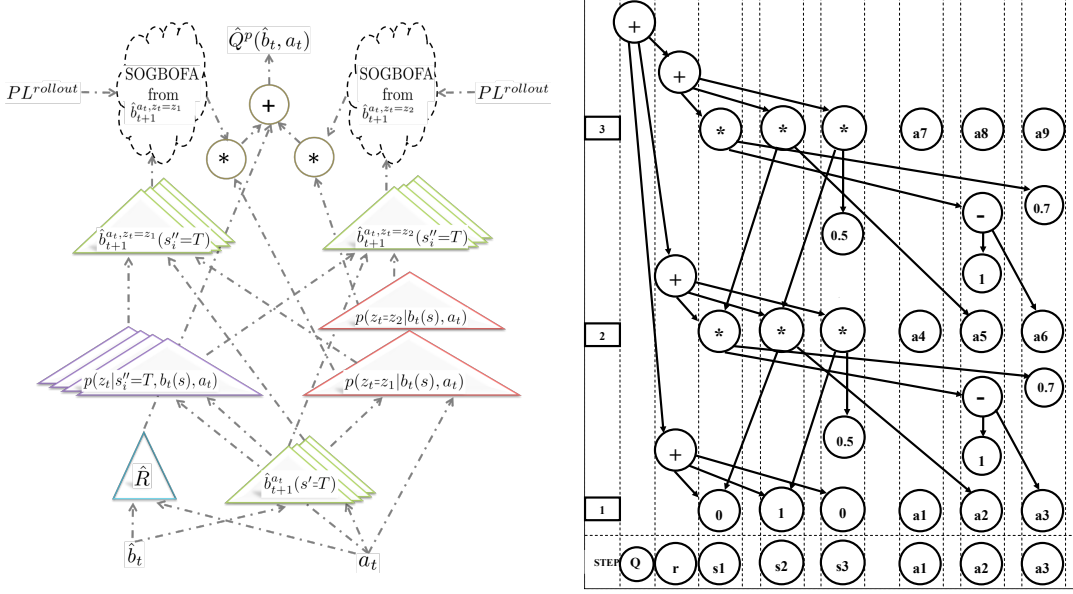

Figure 1: Left: demonstration of the structure of the computation graph in SNAP when there are two possible values for observations $z_t = z_1$ or $z_t = z_2$. Right: demonstration of a three-step simulation in SOGBOFA including the representation of conformant actions.

The basic version of our algorithm enumerates all observations and constructs a computation graph to capture an approximate version of Eq (3) as follows:

$$\hat{Q}^p(\hat{b}_t, a_t) = \hat{R}(\{\hat{b}_t(s_i)\}, a_t) + \sum_{z_t} \left( \prod_k \hat{p}(z_{t,k} | \{\hat{b}_{t+1}^{a_t}(s_i')\}, a_t) \right) \hat{V}^{p^{z_t}}(\hat{b}_{t+1}^{a_t, z_t}). \qquad (7)$$

The overall structure has a sum of the reward portion and the next state portion. The next state portion has a sum over all concrete values for observations. For each concrete observation value we have a product between two portions: the probability for $z_t$ and the approximate future value obtained from $\hat{b}_{t+1}^{a_t, z_t}$. To construct this portion, we first build a graph that computes $\hat{b}_{t+1}^{a_t, z_t}$ and then apply $\hat{V}$ to the belief state which is the output of this graph. This value $\hat{V}^p(b)$ is replaced by the SOGBOFA graph which rolls out $p$ on the belief state. This is done using the computation of $\{\hat{b}_{t+1}^{a_t}(s_i')\}$ which is correct because actions in $p$ are not conditioned on states. As explained above, the computation in SOGBOFA already handles product distributions over state variables so no change is needed for this part. Figure 1 shows the high level structure of the computation graph for POMDPs.

**Example: Tiger Problem:** To illustrate the details of this construction consider the well known Tiger domain with horizon 2, i.e. where the rollout portion is just an estimate of the reward at the second step. In Tiger we have one state variable $L$ (true when tiger is on left), three actions listen, openLeft and openRight, and one observation variable $H$ (relevant on listen; true when we hear noise on left, false when we hear noise on right). If we open the door where the tiger is, the reward is $-100$ and the trajectory ends. If we open the other door where there is gold the reward is $+10$ and the game ends. The cost of taking a listen action is -10. If we listen then we hear noise on the correct side with probability $0.85$ and on the other side with probability $0.15$. The initial belief state is $p(L{=}T) = 0.5$. Note that the state always remains the same in this problem: $p(L'{=}v|L{=}v){=}1$.

We have $p(H{=}T|L', \texttt{listen}) =$ if $L'$ then $0.85$ else $0.15$ which is translated to $L' * 0.85 + (1{-}L') * 0.15$. The reward is $R = ((1{-}L) * \texttt{openRight} + L * \texttt{openLeft}) * -100 + ((1{-}L) * \texttt{openLeft} + L * \texttt{openRight}) * 10 + \texttt{listen} * -10$. We first calculate the approximated $\hat{Q}^p(\hat{b}_t, a_t{=}\texttt{listen})$. The reward expectation of taking the action listen is -10. According to Eq (6), the belief state after hearing noise is $\hat{b}_{t+1}^{a_t{=}\texttt{listen}, H{=}T}(L{=}T) = 0.85$. With the approximation in Eq (4), the reward expectation at step $t + 1$ is then calculated as $E_{\hat{b}_{t+1}^{a_t{=}\texttt{listen}, H{=}T}}[R(s, a_{t+1})] \approx (0.15 * \texttt{openRight}_{t+1}^1 + 0.85 *$

$\mathtt{openLeft}^1_{t+1}) * -100 + (0.15 * \mathtt{openLeft}^1_{t+1} + 0.85 * \mathtt{openRight}^1_{t+1}) * 10 + \mathtt{listen}^1_{t+1} * -10$, where the superscript $o$ of action is to denote that it works with the belief state $o$ after seeing the $o_{th}$ observation. Similarly we have $\hat{b}^{a_t=\mathtt{listen},H=F}_{t+1}(L{=}T) = 0.15$, and the reward expectation on the belief state is calculated as $E_{\hat{b}^{a_t=\mathtt{listen},H=F}_{t+1}}[R(s,a_{t+1})] \approx (0.85 * \mathtt{openRight}^2_{t+1} + 0.15 *$ $\mathtt{openLeft}^2_{t+1}) * -100 + (0.85 * \mathtt{openLeft}^2_{t+1} + 0.15 * \mathtt{openRight}^2_{t+1}) * 10 + \mathtt{listen}^2_{t+1} * -10$. Note that we have $\hat{p}(H{=}T) = \hat{p}(H{=}F) = 0.5$. Now with horizon 2, we have $\hat{Q}^p(\hat{b}_t, a_t = \mathtt{listen}) = -10 + 0.5 * E_{\hat{b}^{a_t=_T\mathtt{listen},H=T}_{t+1}}[R(s,a^1_{t+1})] + 0.5 * E_{\hat{b}^{a_t ceq}_{t+1}\mathtt{listen},H=F}[R(s,a^2_{t+1})]$. Note that the conformant actions for step $t+1$ on different belief states are different. With $\mathtt{openLeft}^1_{t+1}{=}T$ and $\mathtt{openRight}^2_{t+1}{=}T$, the total $Q$ estimate is $-26.5$. Similar computations for $\mathtt{openLeft}$ and $\mathtt{openRight}$ yield $\hat{Q} = -90$. Maximizing over $a_t$ and $p$ we have an optimal conformant path $\mathtt{listen}_t, \mathtt{openLeft}_{t+1}|H{=}T, \mathtt{openRight}_{t+1}|H{=}F$.

# 4 Sampling networks for large observation spaces

The algorithm of the previous section is too slow when there are many observations because we generate a sub-graph of the simulation for every possible value. Like other algorithms, when the observation space is large we can resort to sampling observations and aggregating values only for the observations sampled. Our construction already computes a node in the graph representing an approximation of $p(z_{t,k}|b_t, a_t)$. Therefore we can sample from the product space of observations conditioned on $a_t$. Once a set of observations are sampled we can produce the same type of graph as before, replacing the explicit calculation of expectation with an average over the sample as in the following equation, where $N$ is total number of samples and $z^n_t$ is the $n_{th}$ sampled observation

$$\hat{Q}^p(\hat{b}_t, a_t) = \hat{R}(\{\hat{b}_t(s_i)\}, a_t) + \frac{1}{N}\sum_{n=1}^{N}\hat{V}^{p^{z^n_t}}(\hat{b}^{a_t,z^n_t}_{t+1}). \tag{8}$$

However, to implement this idea we must deal with two difficulties. The first is that during gradient search $a_t$ is not a binary action but instead it represents a product of Bernoulli distributions $\prod_\ell p(a_{t,\ell})$ where each $p(a_{t,\ell})$ determines our choice for setting action variable $a_{t,\ell}$ to true. This is easily dealt with by replacing variables with their expectations as in previous calculations. The second is more complex because of the gradient search. We can correctly sample as above, calculate derivatives and update $\prod_\ell p(a_{t,\ell})$. But once this is done, $a_t$ has changed and the sampled observations no longer reflect $p(z_{t,k}|b_t, a_t)$. The computation graph is still correct, but the observations may not be a

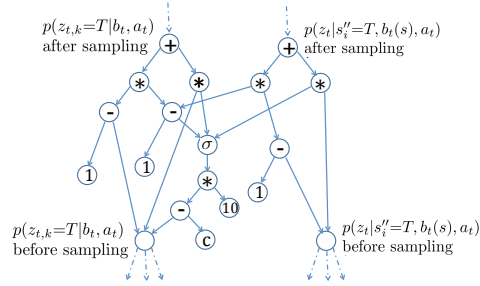

Figure 2: Sampling network structure.

representative sample for the updated action. To address this we introduce the idea of sampling networks. This provides a static construction that samples observations with correct probabilities for any setting of $a_t$. Since we deal with product distributions we can deal with each variable separately. Consider a specific variable $z_{t,k}$ and let $x_1$ be the node in the graph representing $\hat{p}(z_{t,k}{=}T|\{\hat{b}^{a_t}_{t+1}(s'_i)\}, a_t)$. Our algorithm draws $C \in [0,1]$ from the uniform distribution *at graph construction time*. Note that $p(C \le x_1) = x_1$. Therefore we can sample a value for $z_{t,k}$ at construction time by setting $z_{t,k}{=}T$ iff $x_1 - C \ge 0$. To avoid the use of a non-differential condition ($\ge 0$) in our graph we replace this with $\hat{z}_{t,k} = \sigma(A(x_1 - C))$ where $\sigma(x) = 1/(1+e^{-x})$ is the sigmoid function and $A$ is a constant ($A = 10$ in our experiments). This yields a node in the graph representing $\hat{z}_{t,k}$ whose value is $\approx 0$ or $\approx 1$. The only problem is that at graph construction time we do not know whether this value is 0 or 1. We therefore need to modify the portion of the graph that uses $\hat{p}(z_{t,k}|\dots)$ where the construction has two variants of this with different conditioning events, and we use the same solution in both cases. For concreteness let $x_2$ be the node in the graph representing $\hat{p}(z_{t,k}|s''_i{=}T, \{\hat{b}^{a_t}_{t+1}(s''_\ell)\}_{\ell\neq i}, a_t)$. The value computed by node $x_2$ is used as input to other nodes. We replace these inputs with

$$\hat{z}_{t,k} * x_2 + (1 - \hat{z}_{t,k}) * (1 - x_2).$$

Now, when $\hat{z}_{t,k} \approx 1$ we get $x_2$ and when it is $\approx 0$ we get $1 - x_2$ as desired. We use the same construction with $x_1$ to calculate the probability with the second type of conditioning. Therefore, the sampling networks are produced at graph construction time but they produce symbolic nodes representing concrete samples for $z_t$ which are correctly sampled from the distribution conditioned on $a_t$. Figure 2 shows the sampling network for one $\hat{z}_{t,k}$ and the calculation of the probability.

SNAP tests if the observation space is smaller than some fixed constant ($S = 10$ in the experiments). If so it enumerates all observations. Otherwise, it integrates sampling networks for up to $S$ observations into the previous construction to yield a sampled graph. The process for the dynamic setting of simulation depth from SOGBOFA is used for the rollout from all samples. If the algorithm finds that there is insufficient time it generates less than $S$ samples with the goal of achieving at least $n = 200$ gradient updates. Optimization proceeds in the same manner as before with the new graph.

## 5  Discussion

SNAP has two main assumptions or sources of potential limitations. The first is the fact that the rollout plans do not depend on observations beyond the first step. Our approximation is distinct from the QMDP approximation [13] which ignores observations altogether. It is also different from the FIB approximation of [9] which uses observations from the first step but uses a state based approximation thereafter, in contrast with our use of a conformant plan over the factored belief state. The second limitation is the factoring into products of independent variables. Factoring is not new and has been used before for POMDP planning (e.g. [14, 15, 16]) where authors have shown practical success across different problems and some theoretical guarantees. However, the manner in which factoring is used in our algorithm, through symbolic propagation with gradient based optimization, is new and is the main reason for efficiency and improved search.

POMCP [21] and DESPOT [25] perform search in belief space and develop a search tree which optimizes the action at every branch in the tree. Very recently these algorithms were improved to handle large, even continuous, observation spaces [23, 7]. Comparing to these, the rollout portion in SNAP is more limited because we use a single conformant sequential plan (i.e., not a policy) for rollout and do not expand a tree. On the other hand the aggregate simulation in SNAP provides a significant speedup. The other main advantage of SNAP is the fact that it samples and computes its values symbolically because this allows for effective gradient based search in contrast with unstructured sampling of actions in these algorithms. Finally, [21, 25] use a particle based representation of the belief space, whereas SNAP uses a product distribution. These represent different approximations which may work well in different problems.

In terms of limitations, note that deterministic transitions are not necessarily bad for factored representations because a belief focused on one state is both deterministic and factored and this can be preserved by the transition function. For example, the work of [15] has already shown this for the well known *rocksample* domain. The same is true for the T-maze domain of [1]. Simple experiments (details omitted) show that SNAP solves this problem correctly and that it scales better than other systems to large mazes. SNAP can be successful in these problems because one step of observation is sufficient and the reward does not depend in a sensitive manner on correlation among variables.

On the other hand, we can illustrate the limitations of SNAP with two simple domains. The first has 2 states variables $x_1, x_2$, 3 action variables $a_1, a_2, a_3$ and one observation variable $o_1$. The initial belief state is uniform over all 4 assignments which when factored is $b_0 = (0.5, 0.5)$, i.e., $p(x_1 = 1) = 0.5$ and $p(x_2 = 1) = 0.5$. The reward is if $(x_1 == x_2)$ then $1$ else $-1$. The actions $a_1, a_2$ are deterministic where $a_1$ deterministically flips the value of $x_1$, that is: $x_1' =$ if $(a_1 \wedge x_1)$ then $0$ elseif $(a_1 \wedge \bar{x}_1)$ then $1$ else $x_1$. Similarly, $a_2$ deterministically flips the value of $x_2$. The action $a_3$ gives a noisy observation testing if $x_1 == x_2$ as follows: $p(o = 1) =$ if $(a_3 \wedge x_1' \wedge x_2') \vee (a_3 \wedge \bar{x}_1' \wedge \bar{x}_2')$ then $0.9$ elseif $a_3$ then $0.1$ else $0$. In this case, starting with $b_0 = (0.5, 0.5)$ it is obvious that the belief is not changed with $a_1, a_2$ and calculating for $a_3$ we see that $p(x_1' = 1 | o = 1) = \frac{0.5 \cdot 0.9 + 0.5 \cdot 0.1}{(0.5 \cdot 0.9 + 0.5 \cdot 0.1) + (0.5 \cdot 0.9 + 0.5 \cdot 0.1)} = 0.5$ so the belief does not change. In other words we always have the same belief and same expected reward (which is zero) and the search will fail. On the other hand, a particle based representation of the belief state will be able to concentrate on the correct two particles (00,11 or 01,10) using the observations.

The second problem has the same state and action variables, same reward, and $a_1, a_2$ have the same dynamics. We have two sensing actions $a_3$ and $a_4$ and two observation variables. Action $a_3$ gives

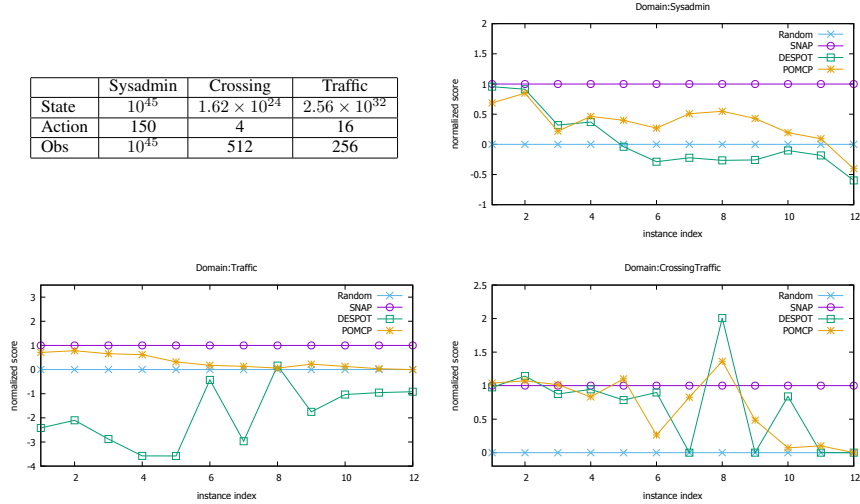

Figure 3: Top Left: The size of state, action, and observation spaces for the three IPC domains. Other Panels: Average reward of algorithms normalized relative to SNAP (score=1) and Random (score=0).

a noisy observation of the value of $x_1$ as follows: $p(o_1 = 1) =$ if $(a_3 \land x_1')$ then $0.9$ elseif $(a_3 \land \bar{x}_1')$ then $0.1$ else $0$. Action $a_4$ does the same w.r.t. $x_2$. In this case the observation from $a_3$ does change the belief, for example: $p(x_1' = 1 | o_1 = 1) = \frac{0.5 \cdot 0.9}{0.5 \cdot 0.9 + 0.5 \cdot 0.1} = 0.9$. That is, if we observe $o_1 = 1$ then the belief is $(0.9, 0.5)$. But the expected reward is still: $0.9 \cdot 0.5 + 0.1 \cdot 0.5 - 0.9 \cdot 0.5 - 0.1 \cdot 0.5 = 0$ so the new belief state is not distinguishable from the original one, *unless one uses additional sensing action $a_4$ to identify the value of $x_2$*. In other words for this problem we must develop a search tree because one level of observations does not suffice. If we were to develop such a tree we can reach belief states like $(0.9, 0.9)$ that identifies the correct action and we can succeed despite factoring, but SNAP will fail because the search is limited to one level of observations. Here too a particle based representation will succeed because it retains the correlation between $x_1, x_2$.

## 6 Experimental evaluation

We compare the performance of SNAP to the state-of-the-art online planners for POMDP. Specifically, we compare to POMCP [21] and DESPOT [25]. For DESPOT, we use the original implementation from https://github.com/AdaCompNUS/despot/. For POMCP we use the implementation from the winner of IPC2011 Boolean POMDP track, POMDPX NUS. POMDPX NUS is a combination of an offline algorithm SARSOP [12] and POMCP. It triggers different algorithms depending on the size of the problem. Here, we only use their POMCP implementation. DESPOT and POMCP are domain independent planners, but previous work has used manually specified domain knowledge to improve their performance in specific domains. Here we test all algorithms without domain knowledge.

We compare the planners on 3 IPC domains. In `CrossingTraffic`, the robot tries to move from one side of a river to the other side, with a penalty at every step when staying in the river. Floating obstacles randomly appear upstream in the river and float downstream. If running into an obstacle, the robot will be trapped and cannot move anymore. The robot has partial observation of whether and where the obstacles appear. The `sysadmin` domain models a network of computers. A computer has a probability of failure which depends on the proportion of all other computers connected to it that are running. The agent can reboot one or more computers, which has a cost but makes sure that the computer is running in the next step. The goal is to keep as many computers as possible running for the entire horizon. The agent has a stochastic observation of whether each computer is still running. In the `traffic` domain, there are multiple traffic lights that work at different intersections of roads. Cars flow from the roads to the intersections and the goal is to minimize the number of cars waiting at intersections. The agent can only observe if there are cars running into each intersection and in which direction but not their number. For each domain, we use the original 10 problems from the competition, but add two larger problems, where, roughly speaking, the problems get harder as their

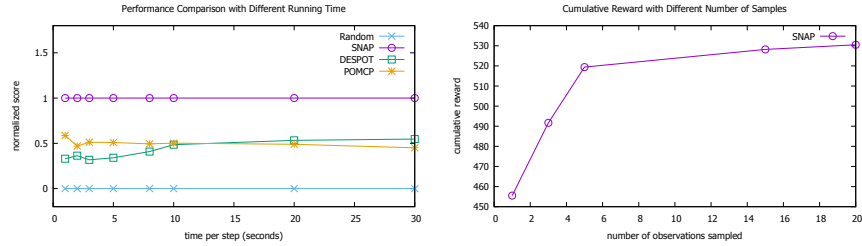

Figure 4: Left: performance analysis of SNAP given different amount of running time. Right: performance analysis of SNAP given different number of sampled observations.

index increases. The size of the largest problem for each domain is shown in Figure 3. Note that the action spaces are relatively small. Similar to SOGBOFA [3], SNAP can handle much larger action spaces whereas we expect POMCP and DESPOT to do less well if the action space increases.

For the main experiment we use 2 seconds planning time per step for all planners. We first show the normalized cumulative reward that each planner gets from 100 runs on each problem. The raw scores for individual problems vary making visualization of results for many problems difficult. For visual clarity of comparison across problems we normalize the total reward of each planner by linear scaling such that SNAP is always 1 and the random policy is always 0. We do not include standard deviations in the plots because it is not clear how to calculate these for normalized ratios. Raw scores and standard deviations of the mean estimate for each problem are given in the supplementary materials. Given these scores, visible differences in the plots are statistically significant so the trends in the plots are indicative of performance. The results are shown in Fig 3. First, we can observe that SNAP has competitive performance on all domains and it is significantly better on most problems. Note that the observation space in sysadmin is large and the basic algorithm would not be able to handle it, showing the importance of sampling networks. Second, we can observe that the larger the problem is, the easier it is to distinguish our planner from the others. This illustrates that SNAP has an advantage in dealing with large combinatorial state, action and observation spaces.

To further analyze the performance of SNAP we explore its sensitivity to the setting of the experiments. First, we compare the planners with different planning time. We arbitrarily picked one of the largest problems, sysadmin 10, for this experiment. We vary the running time from 1 to 30 seconds per step. The results are in Fig 4, left. We observe that SNAP dominates other planners regardless of the running time and that the difference between SNAP and other planners is maintained across the range. Next, we evaluate the sensitivity of SNAP to the number of observation samples. In this experiment, in order to isolate the effect of the number of samples, we fix the values of dynamically set parameters and do not limit the run time of SNAP. In particular we fix the search depth (to 5) and the number of updates (to 200) and repeat the experiment 100 times. The number of observations is varied from 1 to 20. We run the experiments on a relatively small problem, sysadmin 3, to control the run time for the experiment. The results are in right plot of Fig 4. We first observe that on this problem allowing more samples improves the performance of the algorithm. For this problem the improvement is dramatic until 5 samples and from 5 to 20 the improvement is more moderate. This illustrates that more samples are better but also shows the potential of small sample sizes to yield good performance.

## 7 Conclusion

The paper introduces SNAP, a new algorithm for solving POMDPs by using sampling networks and aggregate simulation. The algorithm is not guaranteed to find the optimal solution even if is given unlimited time, because it uses independence assumptions together with inference using belief propagation (through the graph construction) for portions of its computation. On the other hand, as illustrated in the experiments, when time is limited the algorithm provides a good tradeoff as compared to state of the art anytime exact solvers. This allows scaling POMDP solvers to factored domains where state, observation and actions spaces are all large. SNAP performs well across a range of problem domains without the need for domain specific heuristics.

**Acknowledgments**

This work was partly supported by NSF under grant IIS-1616280 and by an Adobe Data Science Research Award. Some of the experiments in this paper were performed on the Tufts Linux Research Cluster supported by Tufts Technology Services.

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
