[Supplementary Material · appendix.pdf]

# Supplementary Matarial for: Sampling Networks and Aggregate Simulation for Online POMDP Planning

**Hao Cui**
Department of Computer Science
Tufts University
Medford, MA 02155, USA
hao.cui@tufts.edu

**Roni Khardon**
Department of Computer Science
Indiana University
Bloomington, IN, USA
rkhardon@iu.edu

## Abstract

The main paper normalizes the cumulative reward obtained by the algorithms in experiments in order to facilitate the visualization across many problems where the scale of reward across problems is different. The supplement gives the raw scores in these experiments.

Table 1: Raw scores in main experiment on the Sysadmin domain.

| sysadmin | SNAP | | Despot | | pomcp | |
|---|---|---|---|---|---|---|
| 1 | 343.03 | 2.643915846 | 337.121 | 30.0229 | 301.188 | 3.564384177 |
| 2 | 332.12 | 4.548654306 | 318.64 | 43.859 | 308.622 | 4.743675069 |
| 3 | 475.87 | 6.906310954 | 375.159 | 63.6787 | 360.37 | 4.999108621 |
| 4 | 420.37 | 7.574650553 | 334.295 | 56.9805 | 347.06 | 4.938493473 |
| 5 | 576.11 | 6.284359872 | 417.942 | 52.7475 | 484.651 | 5.332813047 |
| 6 | 442.7 | 5.567539852 | 323.932 | 41.3483 | 375.262 | 5.368180656 |
| 7 | 655.34 | 7.643012757 | 481.116 | 60.5598 | 585.11 | 6.295544377 |
| 8 | 544.42 | 7.038951342 | 400.552 | 50.8045 | 492.97 | 6.748378748 |
| 9 | 790.18 | 6.909694349 | 565.679 | 76.5311 | 688.33 | 6.901536568 |
| 10 | 873.42 | 8.765171761 | 496.466 | 65.2078 | 597.834 | 5.627034249 |
| 11 | 994.23 | 8.20409477 | 832.149 | 6.8086 | 870.13 | 7.567754819 |
| 12 | 1250.41 | 9.740011242 | 1126.59 | 8.12555 | 1141.69 | 7.437848374 |
| | Random | | Noop | | | |
| 1 | 209.453 | 3.2 | 113.81 | 3.3 | | |
| 2 | 177.333 | 3.1 | 90.99 | 2.7 | | |
| 3 | 327.916 | 5.0 | 224.65 | 4.4 | | |
| 4 | 283.756 | 4.2 | 191.36 | 3.6 | | |
| 5 | 423.961 | 5.6 | 314.48 | 5.6 | | |
| 6 | 350.461 | 5.1 | 255.35 | 4.3 | | |
| 7 | 512.747 | 5.5 | 405.26 | 6.6 | | |
| 8 | 430.727 | 5.0 | 345.12 | 5.1 | | |
| 9 | 611.755 | 7.0 | 497.4 | 6.4 | | |
| 10 | 531.395 | 6.4 | 436.89 | 5.8 | | |
| 11 | 857.296 | 7.4 | 786.99 | 7.2 | | |
| 12 | 1172.911 | 9.2 | 1131.16 | 8.9 | | |

Table 2: Raw Scores in main experiment on the Crossing Traffic domain.

| crossing | SNAP | | Despot | | pomcp | |
|---|---|---|---|---|---|---|
| 1 | -10 | 1.546867803 | -10.74 | 1.59916 | -8.84 | 1.459912326 |
| 2 | -18.72 | 1.886270394 | -15.69 | 18.2328 | -17.2 | 1.861612205 |
| 3 | -16.32 | 1.776 | -19.28 | 1.83663 | -15.97 | 1.763431598 |
| 4 | -26.68 | 1.776 | -27.42 | 17.5272 | -28.91 | 1.647563246 |
| 5 | -14.08 | 1.616395991 | -19.66 | 1.77072 | -11.47 | 1.385817809 |
| 6 | -21.96 | 1.755899769 | -23.8 | 17.9098 | -35.24 | 1.110956345 |
| 7 | -19.7 | 1.727454775 | -40 | 0 | -23.2 | 1.493050568 |
| 8 | -34.77 | 1.245058633 | -29.5 | 16.039 | -32.87 | 1.874328478 |
| 9 | -19.6 | 1.665653025 | -40 | 0 | -30.11 | 1.208378666 |
| 10 | -25.46 | 1.572477027 | -27.79 | 15.6405 | -38.95 | 0.5058408841 |
| 11 | -31.08 | 1.409374329 | -40 | 0 | -39.09 | 0.4552131369 |
| 12 | -32.88 | 1.173139378 | -40 | 0 | -40 | 0 |
| crossing | Random | | Noop | | | |
| 1 | -40 | 0 | -40 | 0 | | |
| 2 | -40 | 0 | -40 | 0 | | |
| 3 | -40 | 0 | -40 | 0 | | |
| 4 | -40 | 0 | -40 | 0 | | |
| 5 | -40 | 0 | -40 | 0 | | |
| 6 | -40 | 0 | -40 | 0 | | |
| 7 | -40 | 0 | -40 | 0 | | |
| 8 | -40 | 0 | -40 | 0 | | |
| 9 | -40 | 0 | -40 | 0 | | |
| 10 | -40 | 0 | -40 | 0 | | |
| 11 | -40 | 0 | -40 | 0 | | |
| 12 | -40 | 0 | -40 | 0 | | |

Table 3: Raw Scores in main experiment on the Traffic domain.

| traffic | SNAP | | Despot | | pomcp | |
|---|---|---|---|---|---|---|
| 1 | -14.71 | 0.425745229 | -88.23 | 1.23027 | -20.96 | 0.6871564596 |
| 2 | -8.31 | 0.3897935351 | -66.58 | 1.87628 | -12.44 | 0.5553953547 |
| 3 | -30.74 | 1.148270003 | -146.31 | 2.43132 | -41.02 | 1.34781156 |
| 4 | -13.07 | 0.6122507656 | -102.09 | 2.91091 | -20.53 | 1.141968038 |
| 5 | -11.45 | 0.8274509049 | -99.0875 | 23.15 | -24.5 | 1.678302714 |
| 6 | -104.71 | 2.174410035 | -195.54 | 3.44393 | -157.07 | 3.717317716 |
| 7 | -36.36 | 1.472923623 | -149.85 | 4.82921 | -61.12 | 2.490352585 |
| 8 | -61.3 | 2.259623863 | -100.868 | 34.3919 | -105.82 | 3.578893125 |
| 9 | -34.07 | 1.154231779 | -141.012 | 29.5179 | -64.26 | 2.914811143 |
| 10 | -31.18 | 1.643008217 | -124.558 | 39.5563 | -71.09 | 3.483621535 |
| 11 | -28.62 | 1.094839474 | -132.345 | 38.9843 | -79.854 | 3.374344783 |
| 12 | -29.132 | 2.049343448 | -134.938 | 34.4459 | -84.434 | 3.343453555 |
| traffic | Random | | Noop | | | |
| 1 | -36.2 | 1.371641353 | -75.32 | 0.5425642819 | | |
| 2 | -27.1 | 1.308930861 | -52.44 | 1.530707026 | | |
| 3 | -60.53 | 2.298192986 | -166.74 | 1.542440923 | | |
| 4 | -32.52 | 1.619226976 | -92.34 | 2.23012197 | | |
| 5 | -30.59 | 1.856830364 | -113.98 | 3.752651862 | | |
| 6 | -168.22 | 4.459968161 | -224.6 | 2.123110925 | | |
| 7 | -64.95 | 2.539896651 | -241.27 | 3.625930363 | | |
| 8 | -108.59 | 3.8802344 | -282.66 | 3.042243251 | | |
| 9 | -72.93 | 3.308179409 | -263.44 | 4.036070366 | | |
| 10 | -77.07 | 3.42082607 | -246.88 | 3.50799886 | | |
| 11 | -81.68 | 4.439554736 | | | | |
| 12 | -84.32 | 4.780721092 | | | | |

Table 4: Raw scores for the three algorithms when varying the time per step (in seconds) on Sysadmin problem 10.

| time | SNAP | | POMCP | | Despot | |
|------|----------|---------|----------|--------|----------|---------|
| 1    | 754.5499 | 4.7794  | 578.3649 | 2.22577| 468.655  | 2.0867  |
| 2    | 852.4999 | 4.7726  | 574.639  | 2.5382 | 518.94   | 2.8497  |
| 3    | 872.249  | 6.448   | 606.9195 | 2.9298 | 500.944  | 1.39138 |
| 5    | 870.6499 | 4.197   | 604.1599 | 2.617  | 512.673  | 1.2927  |
| 8    | 877.549  | 5.05086 | 599.4049 | 2.9788 | 552.82   | 3.7258  |
| 10   | 887.999  | 4.85948 | 608.9849 | 2.0722 | 599.54   | 3.9667  |
| 20   | 880.9999 | 4.7474  | 598.714  | 2.993  | 622.714  | 3.12378 |
| 30   | 920.099  | 4.85697 | 594.9766 | 2.2836 | 651.980  | 2.3241  |

Table 5: Raw scores for SNAP when varying the number of sampled observations on Sysadmin problem 3. In order to isolate the effect of the number of samples in this experiment, the time per step is not limited, the graph depth is fixed to 5, and the number of updates is fixed to 200.

| #samples | Average reward | Standard deviation |
|----------|----------------|--------------------|
| 1        | 455.5          | 1.832170844        |
| 3        | 491.7          | 2.263297273        |
| 5        | 519.4          | 2.344324343        |
| 15       | 528.2          | 2.203896549        |
| 20       | 530.5          | 2.048572674        |