[Reviews · NeurIPS 2019]

Reviewer 1



Author feedback: I thank the authors for the feedback. The feedback was of high quality and satisfied my concerns. I suggest that, perhaps a compressed version, of "Explaining limitations of our work" from the author feedback, which I enjoyed reading, will be added to the final version of the paper. The paper "Sampling Networks and Aggregate Simulation for Online POMDP Planning" proposes a new solution to computing policies for large POMDP problems that is based on factorizing the belief distribution using a mean field approximation during planning and execution and extending aggregate simulation to POMDPs. In short, the proposed POMDP planner projects factorized beliefs forward in time forming at the same time a computational graph and then computes gradients backwards in time over the graph to improve the policy. Overall, the approach seems promising and outperforms state-of-the-art in large problems where the approximations do not seem to degrade performance. The approach relies on a factorized approximation which is not accurate contrary to particle based approximation which are accurate in the limit. Due to this the experiments should be extended with more classic POMDP problems, for example the RockSample problem where the factorization is suspected to not work well. This would give a more complete picture of the strengths and weaknesses of the approach. I would like to emphasize that even though the proposed approach may not work in all tasks it provides a novel approach to POMDP solving and could be beneficial in many practical problems. EXPERIMENTS: "Here we test all algorithms without domain knowledge." It is well known that MCTS based algorithms (such as POMCP) only work well with a suitable rollout heuristic (the heuristic can also be learned, see e.g. Alpha Go Zero). Also results with a heuristic (domain knowledge) even if with a simple one should be provided. It can be also considered "fair" to provide some domain knowledge to MCTS approaches in the form of a rollout heuristic since adding domain knowledge is straightforward compared to other approaches. Figures 3 and 4 should include confidence intervals. RELATED WORK: Using factored beliefs in POMDP planning is not new. For example, [Pajarinen et al., 2010] uses factored beliefs to solve large POMDP problems. These results also give hints on when a factored belief may be problematic. For example, in the classic RockSample problem, where movements are deterministic, a factored probability distribution does not seem to be a good approximation. Pajarinen, J., Peltonen, J., Hottinen, A., & Uusitalo, M. A. (2010). Efficient planning in large POMDPs through policy graph based factorized approximations. In Joint European Conference on Machine Learning and Knowledge Discovery in Databases (pp. 1-16). Springer, Berlin, Heidelberg. LANGUAGE: "A MDP" -> "An MDP" "each of these is specific by a" -> "each of these is specified by a" "it is important to noice" -> "it is important to notice" "which rolls out out" -> "which rolls out" "shows the potential of sample sample sizes to yield good performance" ?? In Figure 1, there is the text "SOGBOA". Should this be "SOGBOFA"?

Reviewer 2



The paper extends the previous H. Cui et al. work [2,3,4] to POMDP problems. While the previous method factorizes the state space, the authors consider factoring over the representation of belief state. A symbolic computation graph of the value function is introduced and is used for optimizing the policy by gradient. The numerical experiments show better performance than the state-of-the-art baselines. Strengths are as written in 1 Contributions. Weaknesses: - There is no theoretical analysis. - The numerical evaluation might be limited since the tasks used in the experiments seems to be biased. While these tasks have a large number of states, they might have small impact on the technical assumptions, especially the factorization over the representation of belief state, compared to the classical POMDP tasks like the T-maze task [Bakker, 2002]. - Since the proposed method employs several heuristics, which are used in SOGBOFA [2], it is unclear which heuristics contributed significantly to performance in the numerical experiments. Minor issue: - Line 96, p(x=1) should be p(x=T), p(i=T), or something? Also, the notation of T will be missing. At first, I misunderstood it as the time step T. Bakker, B.: Reinforcement learning with long short-term memory. In: Advances in Neural Information Processing Systems, vol. 14. MIT Press (2002) === Update after the rebuttal === I would like to thank the authors for submitting a response to the reviews and addressing my concerns. The explanation of limitations of the proposed approach in the response is interesting and important. I would like to recommend to include the explanation in the main body of the final version.

Reviewer 3



This paper proposes a new approximate POMDP planning algorithm with a few tricks: approximating beliefs using product distributions, sampling observations, and aggregate simulation. The approach enables action selection using gradient-based optimization. The proposed algorithm performs competitively against POMCP and DESPOT on three large IPC problems. The ideas are interesting, and seems to work effectively on some large problems that current algorithms do not scale up. One main limitation is that the method has no guarantee of finding out the optimal policy even if the search time is unlimited though. While POMCP and DESPOT can find optimal policies given sufficient time and performs much better with some prior knowledge, the proposed approach sacrifices performance guanrantee, but seems to perform better without prior knowledge. Does the approximations used in the algorithm have a big impact on the best possible solution found by the algorithm? Comparing the algorithm with other POMDP planning algorithms on smaller datasets can be helpful. Minor comments - Eq. (2): b_{t}(s) should be b_{t}. - Line 88: why is a sequential plan p not a policy? - Line 96: not clear why p_{i} is p(x=1), and it is confusing to use p for different quantities (plan, probability). * Update after the rebuttal The authors provided some examples to illustrate the limitations of the proposed algorithm. They stated that "DESPOT only runs with POMDPX". This is not true. DESPOT can work with a model implemented in C++ as well. So their additional result need to be updated to take this into account as well.

[Author Response · NeurIPS 2019]

We thank the reviewers for their comments. We will address all style suggestions and minor points.

**Explaining limitations of our approach:** The main comment in all 3 reviews is the suggestion to include more discussion or experiments with small problems to illustrate the limitations of the approach, which we agree is a good idea. First note that the main assumptions of SNAP are the independence assumption in rollouts and the fact that the rollout plans do not depend on observations beyond the first step. This last restriction can be lifted at the cost of being exponential in depth, as in other algorithms, but as experiments show speedup is crucial in large problems.

Deterministic transitions are not necessarily bad for factored representations because a belief focused on one state is both deterministic and factored and this can be preserved by the transition function. Both the T-maze and the rocksample domains that were proposed in the reviews are actually suitable for SNAP. The reason is that one step of observation is sufficient and the reward does not depend in a sensitive manner on correlation among variables. The [Pajarinen] paper shows that factoring works well for rocksample. We encoded the T-maze domain in RDDL and show the results in the table. We use 3 seconds per step for each algorithm, and the T-maze length is shown in the first row. YES means that the algorithm finds optimal actions for all steps, NO means the algorithm does not, and X means the result is not available as DESPOT only runs with pomdpx and the

| | 5 | 10 | 12 | 15 | 30 | 50 |
|---|---|---|---|---|---|---|
| SNAP | YES | YES | YES | YES | YES | YES | YES |
| DESPOT | YES | YES | YES | X | X | X | X |
| POMCP | YES | NO | NO | NO | NO | NO | NO |

RDDL to pomdpx translator failed for this problem size. The result shows that SNAP can solve the T-maze problems.

However, we can illustrate the tradeoff with two other simple domains. The first has 2 states variables $x_1, x_2$, 3 action variables $a_1, a_2, a_3$ and one observation variable $o_1$. The initial belief state is uniform over all 4 assignments which when factored is $b_0 = (0.5, 0.5)$, i.e., $p(x_1 = 1) = 0.5$ and $p(x_2 = 1) = 0.5$. The reward is if $(x_1 == x_2)$ then 1 else $-1$. The actions $a_1, a_2$ are deterministic where $a_1$ deterministically flips the value of $x_1$, that is: $x_1' =$ if $(a_1 \wedge x_1)$ then 0 elseif $(a_1 \wedge \bar{x}_1)$ then 1 else $x_1$. Similarly, $a_2$ deterministically flips the value of $x_2$. The action $a_3$ gives a noisy observation testing if $x_1 == x_2$ as follows: $p(o = 1) =$ if $(a_3 \wedge x_1' \wedge x_2') \vee (a_3 \wedge \bar{x}_1' \wedge \bar{x}_2')$ then 0.9 elseif $a_3$ then 0.1 else 0. In this case, starting with $b_0 = (0.5, 0.5)$ it is obvious that the belief is not changed with $a_1, a_2$ and calculating for $a_3$ we see that $p(x_1' = 1|o = 1) = \frac{0.5 \cdot 0.9 + 0.5 \cdot 0.1}{(0.5 \cdot 0.9 + 0.5 \cdot 0.1) + (0.5 \cdot 0.9 + 0.5 \cdot 0.1)} = 0.5$ so the belief does not change. In other words we always have the same belief and same expected reward (which is zero). Therefore, for this problem factoring implies that the search is blind. On the other hand, a particle based representation of the belief state will be able to concentrate on the correct two particles (00,11 or 01,10) using the observations.

The second problem has the same state and action variables, same reward, and $a_1, a_2$ have the same dynamics. We have two sensing actions $a_3$ and $a_4$ and two observation variables. Action $a_3$ gives a noisy observation of the value of $x_1$ as follows: $p(o_1 = 1) =$ if $(a_3 \wedge x_1')$ then 0.9 elseif $(a_3 \wedge \bar{x}_1')$ then 0.1 else 0. Action $a_4$ does the same w.r.t. $x_2$. In this case the observation from $a_3$ does change the belief, for example: $p(x_1' = 1|o_1 = 1) = \frac{0.5 \cdot 0.9}{0.5 \cdot 0.9 + 0.5 \cdot 0.1} = 0.9$. That is, if we observe $o_1 = 1$ then the belief is $(0.9, 0.5)$. But the expected reward is still: $0.9 \cdot 0.5 + 0.1 \cdot 0.5 - 0.9 \cdot 0.5 - 0.1 \cdot 0.5 = 0$ so the new belief state is not distinguishable from the original one, *unless one uses additional sensing action $a_4$ to identify the value of $x_2$*. In other words for this problem we must develop a search tree because one level of observations does not suffice. If we were to develop such a tree we can reach belief states like $(0.9, 0.9)$ that identifies the correct action and we can succeed despite factoring, but SNAP will fail because the search is limited to one level of observations. Here too a particle based representation will succeed because it retains the correlation between $x_1, x_2$.

**Rev1 - Heuristic domain knowledge:** we agree that the performance of MCTS will improve with domain specific knowledge. However, our focus was on domain independent performance of the planners. In addition, since different algorithms might use domain knowledge in a different manner the comparison of algorithms would be less clear.

**Rev1 - Prior work with factored belief:** Thanks! We will add a discussion of this and other work to the paper.

**Rev2 - Evaluating contributions from SOGBOFA:** We agree that it is important to understand the contribution of different components. Contributions of components that are part of SOGBOFA were reported in the cited papers, albeit in the context of MDPs. The experimental evaluation in this paper is centered on evaluating the the new ideas in this paper, showing the importance of sampling for large observation spaces, and performance sensitivity w.r.t. run time per step and number of samples.

**Rev2 and Rev3: line 96:** Yes $p(x = 1)$ should be $p(x = T)$. We will plan to improve the description. In the current notation, for a generic variable $x$ assume $p(x = T)$ is given by the RDDL expression if $(cond1)$ then $p_1$ elseif $(cond2)$ then $p_2$ else $p_3$ where the conditions form a set of mutually exclusive and exhaustive set of conjunctions. Then the probability of $x$ given that $cond_i$ holds is $p_i$. In general, assuming discrete variables, the CPT of the variable $x$ can be rewritten in this form and this is facilitated by the RDDL representation.

**Rev3 - plan vs. policy:** We will plan to improve the description. By a plan we meant a pre-determined sequence of actions not conditioned on states or observations. The same notion is also known as an open loop policy and as a straight line plan. A policy will condition future action choices on the future states (in MDP) or belief states (in POMDP).

[Meta-Review · NeurIPS 2019]

All reviewers appreciate a practical approach to tackle POMDP in large state and observation space with factorized belief and aggregated simulation. Reviewers had some concern regarding the limitation of the work by the factorization assumption, but these concerns are addressed in author feedback. Reviewers are particularly happy about the quality of the rebuttal and encourage authors to incorporate the discussion of limitation of the algorithm in final draft.